computational mechanics/computer modelling and simulation

intracranial pressure, tumour, CSF, deformation, finite-element modelling

# How growing tumour impacts intracranial pressure and deformation mechanics of brain

Ali Ahmed[1], Muhammad Uzair UlHaq[1], Zartasha Mustansar[2], Arslan Shaukat[3] and Lee Margetts[4]

[1]Department of Biomedical Engineering and Sciences, School of Mechanical and Manufacturing Engineering (SMME), [2]Department of Computational Engineering, Research Center of Modeling and Simulation (RCMS), and [3]Department of Computer and Software Engineering, College of Electrical and Mechanical Engineering, National university of Sciences and Technology (NUST), Islamabad 44000, Pakistan
[4]Department of Mechanical, Aerospace and Civil Engineering, University of Manchester, UK

AA, 0000-0002-8661-2758; MUU, 0000-0001-9660-8982; ZM, 0000-0002-2327-7577

**Author for correspondence:**
Zartasha Mustansar
e-mail: zmustansar@rcms.nust.edu.pk

Brain is an actuator for control and coordination. When a pathology arises in cranium, it may leave a degenerative, disfiguring and destabilizing impact on brain physiology. However, the leading consequences of the same may vary from case to case. Tumour, in this context, is a special type of pathology which deforms brain parenchyma permanently. From translational perspective, deformation mechanics and pressures, specifically the intracranial cerebral pressure (ICP) in a tumour-housed brain, have not been addressed holistically in literature. This is an important area to investigate in neuropathy prognosis. To address this, we aim to solve the pressure mystery in a tumour-based brain in this study and present a fairly workable methodology. Using image-based finite-element modelling, we reconstruct a tumour-based brain and probe resulting deformations and pressures (ICP). Tumour is grown by dilating the voxel region by 16 and 30 mm uniformly. Cumulatively three cases are studied including an existing stage of the tumour. Pressures of cerebrospinal fluid due to its flow inside the ventricle region are also provided to make the model anatomically realistic. Comparison of obtained results unequivocally shows that as the tumour region increases its area and size, deformation pattern changes extensively and spreads throughout the brain volume with a greater concentration in tumour vicinity. Second, we conclude that ICP pressures inside the cranium do increase substantially; however, they still remain under the

normal values (15 mmHg). In the end, a correlation relationship of ICP mechanics and tumour is addressed. From a diagnostic purpose, this result also explains why generally a tumour in its initial stage does not show symptoms because the required ICP threshold has not been crossed. We finally conclude that even at low ICP values, substantial deformation progression inside the cranium is possible. This may result in plastic deformation, midline shift etc. in the brain.

# 1. Introduction

A unique situation in human brain may occur wherein the presence of any pathology inside the cranium, such as tumour, may cause one of the intracranial compartments, i.e. brain, blood or cerebrospinal fluid (CSF), to undergo volume changes. These changes mostly result in the compression of the compartment, thereby increasing net intracranial cerebral pressures (ICP). It is well established that the normal ranges of ICP lie in 5–15 mmHg [1]. Management of this value of ICP is an important biomarker in intracranial surgical studies. However, continuous monitoring is cumbersome and risky due to the invasive procedure associated therewith. Furthermore, when analysing and diagnosing various pathologies and treating traumatic brain injuries (TBI), ICP is the most important control variable which needs medical attention [2,3].

Net ICP changes inside the cranium are related to three factors. First, is the mean arterial pressure (MAP) or the blood pressure due to the blood flowing into the brain. Second, is the CSF pressure inside brain ventricles and in subarachnoid space (SAS). CSF inside the cranium ensures the protection of brain from any unwarranted injury [4,5]. It also provides the buoyancy which makes brain float inside the cranium thereby effectively reducing the net brain weight [5]. The third factor comes from the brain volume itself, as it is incompressible. Therefore, if the mass of the brain increases (as it does in cases of haematoma, cerebral oedema or brain tumour), then that compresses the other two intracranial compartments and increases pressures of those compartments. For example, a growing tumour in brainstem may displace the CSF down the spinal canal and may also block CSF coming from the cerebral aqueduct which resultantly increases pressures on brain parenchyma causing permanent deformation, ischaemia or herniation. The combined equation relating ICP and MAP is given as follows:

$$CPP = MAP - ICP, \qquad (1.1)$$

where CPP is called cerebral perfusion pressure (the pressure differential with which blood forces inside the brain). ICP pressure also contains the CSF pressures and the venous drainage pressure. ICP is most sensitive to the CSF pressures; for example, in cases of severe head injuries [6], CSF pressure contributes one-third rise in ICP. Furthermore, greater compliance in terms of pressure relief comes from CSF demotion or drainage through arachnoid villi and extracranial lymphatics [7]. Whereas, as noted earlier, ICP can rise due to multiple reasons. While studying its relationship with any pathology, ICP has a distinct and clinically distinguishing correlation with the type of pathology. Such correlation, if found out without using invasive procedure, has a huge translational merit as it will not only bypass the need to use an invasive procedure but would also improve prognosis of the disease considerably.

Tumour is a type of pathology whose relationship with the ICP has not been fully explored let alone reported in terms of any non-invasive measurement correlational value. The pathophysiological characteristic of a tumour is that its cells multiply aggressively which, when reached at metastasis stage, quickly spreads to other parts of the body. Consequently, we aim to show and model fundamentally:

1. What is the relationship of ICP and tumour particularly for early stage tumours? This will help us adopt functional parameters to reduce risk to the patient's life quantitatively.
2. How a growing tumour affects the physiology of brain especially in its vicinity; specially when the expansion of brain compartments is haphazard.
3. And lastly, is it possible to transform (2) into a beneficial translational measure so that it can be used as an additional tool in diagnosis.

To model the aforementioned aspects, an image-based finite-element model has to be used wherein different biomechanical aspects can be studied. Therefore, a micro-structurally faithful three-dimensional FEM head model is developed considering important brain layers which may induce any

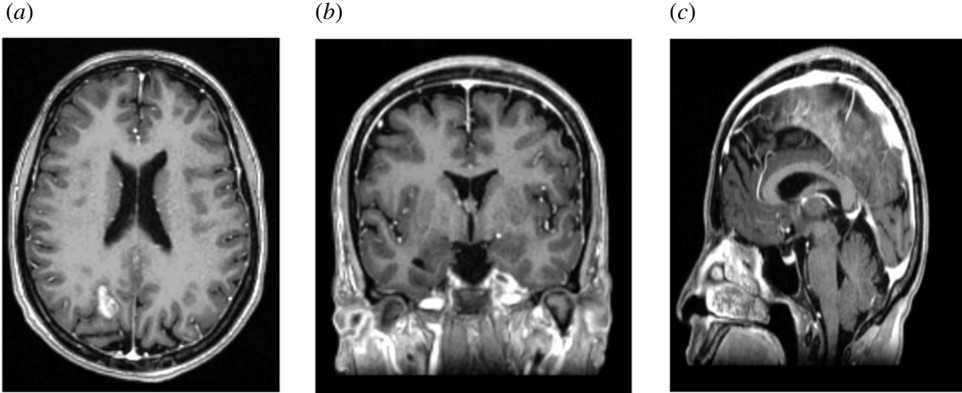

**Figure 1.** MRI image views which shows brain parenchyma and brain tumour region; (*a*) shows axial view and tumour region also; (*b*) shows coronal view and (*c*) shows a sagittal cross-sectional view.

effect. Tumour is grown artificially using the image-based voxel dilation method. By applying the boundary conditions (discussed later), a biomechanical response of brain is calculated which would help in making appropriate conclusions and fill the missing links. Lastly, before moving towards the methods section, we would like to emphasize the need for present study in the current research stream. To assess the gravity of the situation, one can only look for the recent statistics published by Central Brain Tumor Registry of the United States (CBTRUS) documenting yearly cases of brain tumour instances in the US [8]. The said report states that from 2001 to 2015, the 5-year survival rate for patients diagnosed with malignant tumour was only 8% from diagnosed 34.8% and 37.1% males and females, respectively. Hence modelling ICP biomechanics is absolutely important not just for the cases of tumours but also for other relatable situations. This is also especially useful in post-resection of tumour cases wherein resurgence of elevated ICP is likely due to either bleeding in removing tumour bed or due to oedema caused by tumours [9].

# 2. Material and methods

In this study, we have used a real MRI image of a 44-year-old female patient suffering from brain tumour. The initial location of tumour is noted in left anterior parietal lobe (figure 1). While its focal size is 6 mm, other small vasogenic oedemas are also observed; however, for the purposes of this study, these are not considered in the analysis. According to the opinion of radiologist, the tumour is in the early phases of metastasis. Tumour, ventricles, brain, CSF, dura mater and skull are segmented. Segmented masks are three-dimensionally reconstructed and exported as STL file formats. These are later used in ANSYS for analysis. Since in this investigation we want to observe the effect of tumour growth in terms of its size on the deformation mechanics, we have manually grown tumour region by 16 and 30 pixels uniformly using morphological operation called pixel/voxel dilation technique [10]. To the best of our knowledge, there is no cavil to the aforementioned assumption since this investigation addresses some of the fundamental hypotheses, and detailed level studies can be done thereafter once the initial set of hypotheses are tested. Figure 1 shows the MRI image in axial, sagittal and coronal plane of the patient.

## 2.1. Three-dimensional geometry

To segment different brain layers mentioned above, a three-dimensional slicer is used. Thresholding and region growing technique is used to obtain segmented masks. The segmented slices are three-dimensionally reconstructed using Delaunay triangulation scheme [11]. Laplacian smoothing filter is applied to remove any spurious edges or faces. Figure 2*a–d*) shows the three-dimensionally reconstructed regions of brain including the meshing quality, whereas figure 3*a–c* shows original and dilated tumour region by 16 and 30 pixels.

## 2.2. Model meshing

Intricate biological geometries need extra wisdom. Here, we have meshed ventricle (which is a surface body) with SHELL 181 elements at 0.1 mm size. Quadrilateral meshing is done on the surface body

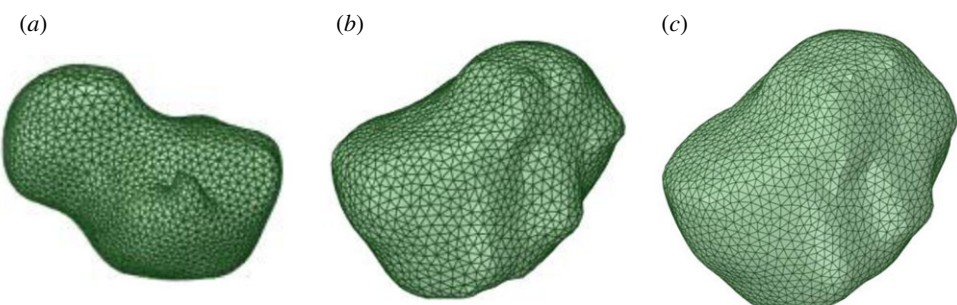

(a)

(b)

(c) skull layer / dura layer / CSF layer / brain layer

(d)

**Figure 2.** Geometry and FEM mesh of the model. Part (*a*) shows the transverse section of the brain model; (*b*) shows the labelling of the regions; (*c*) shows a finer mesh on the ventricle body which is provided for changing loads coming from CSF flow and (*d*) shows the complete mesh model which includes skull and dura mater as well.

**Figure 3.** Tumour regions reconstructed; (*a*) original, (*b*) 16 pixel grown and (*c*) tumour grown by 30 pixels.

using shell elements. Whereas, on other parts such as skull, brain parenchyma etc. SOLID187 elements are used for meshing with a finer size in areas where curvature is present. As far as possible, conformal meshing is used; however, wherever necessary a finer mesh is also used to resolve the stress irregularities. Furthermore, minimum mesh size for the solid parts is 2 mm where the finer mesh is not needed and for finer mesh refinement layer is also used. Parts are meshed in ANSYS mechanical. Due to abrupt and changing loads of ventricle body, a finer mesh is adopted on its surface, so as to resolve stress singularities and overcome convergence issues. This can be visualized in figure 2*c*.

## 2.3. Tissue-level material modelling

Choosing correct material properties to represent soft brain tissue is critical in the analysis. Most material models for brain parenchyma and ventricles are different in literature. It is well established that brain has a nonlinear behaviour under different loading conditions [12]. For instance, some papers in the literature have considered brain parenchyma as linear viscoelastic [12–14], while others have considered brain as hyperelastic [15–18]. Additionally, some papers consider brain parenchyma with both hyperelastic and viscoelastic behaviours [19–21]. Furthermore, recent evidence suggests that soft tissues, such as brain parenchyma and gliomas have unusual mechanical properties especially under shear load [22–24]. According to a paper by Mihai *et al.* [25], this unusual behaviour of brain under shear loading criteria is addressed if one considers hyperelastic models of either *Ogden* or *Mooney–Rivlin* to model brain parenchyma. Similarly, in some papers concerning the area of TBI, the brain parenchyma is also modelled using both viscoelastic and hyperelastic.

Hence, in view of the foregoing, we can safely draw a conclusion that to model brain parenchyma, one needs to exploit both the hyperelastic and viscoelastic (quasi-linear viscoelastic properties propounded by [26]) properties of tissue. Resultantly in this paper, we model brain parenchyma as hyper-viscoelastic. To model it as hyperelastic, a strain density function needs to be calculated which is basically the stored strain energy per unit volume. Using finite strain theory, strain energy density function can be calculated.

Without going into much detail, a hyperelastic model presupposes a function called the strain energy density function (also called Helmholtz free energy per unit volume) denoted by $\varphi$, which is only dependent on the deformation gradient $F$ of the body [27]. So effectively it can be said that $\varphi = \varphi(F)$. $F$ can be calculated as follows [28,29]:

$$F = \frac{\partial x(X, t)}{\partial t}. \tag{2.1}$$

Further, the stress tensor and the Jacobian of the body can be calculated as follows:

$$J = \det[F] \tag{2.2}$$

and

$$\sigma = J^{-1} \frac{\partial \varphi(F)}{\partial F} F^T, \tag{2.3}$$

where $\sigma$ is the Cauchy stress tensor, $\varphi$ is the strain energy and $J$ is the Jacobian of the body as defined in equation (2.2). Based on equations (2.1), (2.2) and (2.3), we now move towards the calculation of strain energy density function, denoted by $\varphi$. The function $\varphi$ is material behaviour dependent. Usually, the strain energy density function is represented/formulated in terms of its principal stretches. Ogden model [30] is formulated in terms of principal invariants or Eigenvectors $\lambda_1$, $\lambda_2$, $\lambda_3$ and the associated strain energy behaviour widely used in modelling biological tissues is given by

$$\varphi = \varphi(\lambda_1, \lambda_2, \lambda_3) = \sum_{n=1}^{N} \frac{\mu_n}{\alpha_n} (\lambda_1^{\alpha_n}, \lambda_2^{\alpha_n}, \lambda_3^{\alpha_n} - 3), \tag{2.4}$$

where $N$ is the number of terms used in determining strain energy density function (such as Ogden two- or three-parameter model), $\mu_n$ are material constants (shear moduli) and $\alpha_n$ are the experimental value (dimensionless).

Whereas the viscoelastic property of the brain parenchyma is calculated using the *generalized standard linear model* and approximated using *Prony series shear relaxation* which is given by

$$G(t) = G_\infty + (G_0 - G_\infty) e^{-(t/\zeta)}, \tag{2.5}$$

where $G_\infty$ is the long-term relaxed shear modulus and $G_0$ is the instantaneous shear modulus, which would be high. Brain tumour is also modelled with the same properties as that of brain parenchyma but with an additional shear stress due to increased stiffness of tumour core [31]. Ventricular body is modelled viscoelastically and is consistent with the literature available on it [32,33]. The material properties are listed in table 1. Figure 4a shows graph of shear relaxation curve using above-mentioned parameters against $\beta = 100 \, \text{s}^{-1}$. Whereas figure 4b shows stress–strain relationship (plot of logarithm stress against strain) of hyperelastic Ogden two-parameter model being used to model brain parenchyma.

**Table 1.** Material properties.

| tissue layer | density (kg m$^{-13}$) | Poisson's ratio | elasticity modulus | Ogden hyperelastic model parameters | viscoelastic model (Prony series constants) | references |
|---|---|---|---|---|---|---|
| skull | 1210 | 0.22 | 8.0 GPa | — | — | Yang et al. [34] |
| ventricles | 1000 | 0.49 | 30 KPa | — | $G_r = 0.0099$, $\beta = 100\ s^{-1}$ | Yang et al. [34] and Masoumi et al. [35] |
| dura mater | 1133 | 0.45 | 31.5 MPa | — | — | Yang et al. [34] |
| brain parenchyma | 1040 | 0.4996 | — | $\mu_1 = 53.8$ Pa, $\alpha_1 = 10.9$, $\mu_2 = -120.4$, $\alpha_2 = -12.9$ | $G_0 = 41$ kPa. $G_\infty = 7.8$ kPa. $\beta = 100\ s^{-1}$ | Ratajczak et al. [36] and Yang et al. [34] |
| CSF | 1000 | 0.49 | bulk modulus = 2.19 GPa | — | — | Yang et al. [34] |
| tumour | 1040 | 0.4996 | shear response | the same as brain parenchyma | the same as brain parenchyma | — |

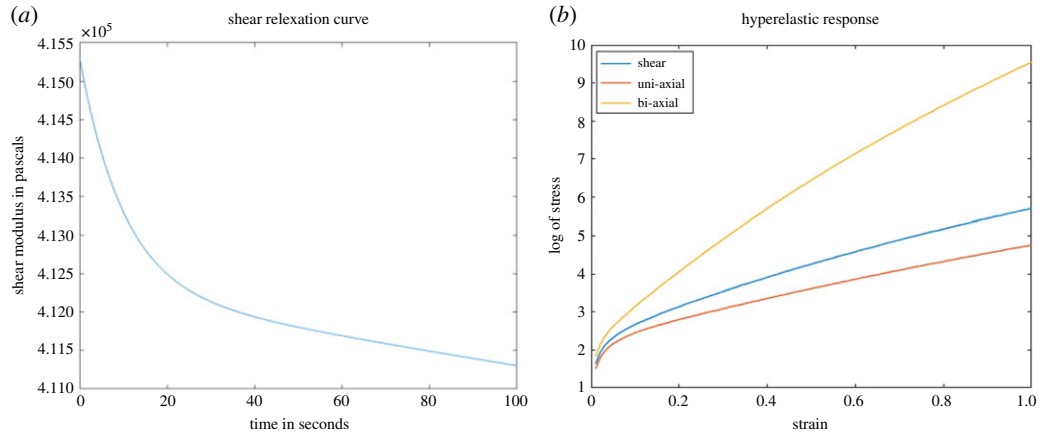

**Figure 4.** These two figures are the material response plots coming from viscous behaviour approximating (*a*) shear relaxation using Prony series and (*b*) shows the Ogden two-parameter hyperelastic model estimation using the properties used in table 1.

**Table 2.** Inter-tissue connectivity and contact formulation scheme.

| tissue connectivity | type of contact | contact formulation scheme | justification |
| --- | --- | --- | --- |
| skull and dura mater | bonded | MPC | Preferred for bonded type contacts. Also gives efficient convergence. |
| CSF and brain parenchyma | frictional ($\mu = 0.2$) | augmented Lagrange formulation | Preferred for frictional contact schemes with large deformation problems. |
| ventricles and brain parenchyma | bonded | MPC | Highly preferred for shell–solid body interface need efficient contact scheme where convergence difficulties can be overcome with no penetration. |
| tumour and brain parenchyma | bonded | MPC | Preferred for bonded type contacts. Also gives efficient convergence. |
| CSF and dura mater | no-separation | Lagrange multiplier formulation | Since no-separation is used and at the same time penetration is to be avoided. |

## 2.4. Boundary conditions

The skull is fixed in space for translation and rotations. Inter-tissue contact definition is provided as follows: skull and dura mater have zero relative motion, therefore they are bonded together with *multipoint constrain formulation* (MPC). Contact definition between CSF layer and brain parenchyma is defined to be *frictional contact* with frictional coefficient $\mu = 0.2$ [37] with *augmented Lagrange formulation*. Ventricles and brain parenchyma have *bonded contact* with MPC formulation. A similar contact is also defined for brain parenchyma and tumour. Dura and CSF layers have contact definition of no-separation with *Lagrange multiplier formulation*. Brief summary of contact definitions is presented in table 2.

Tumour pressure is applied uniformly in radial direction across its body. Since we wanted to make FEM model more realistic, CSF pressures due to its normal flow inside ventricles is applied throughout its surface, normal to the wall. These pressures are numerically calculated using FLUENT against following boundary conditions: inlets and outlets of ventricles are also fixed to constraint the system properly; the surface of ventricle wall is defined over which CSF pressures are transferred; mass flow at the inlet is used with bulk production of 0.3 ml min$^{-1}$ [38]. CSF, taken as Newtonian fluid with a constant viscosity of 0.001003 Pa s, is used [39]. Whereas the density of CSF fluid is taken to be 1000 kg m$^{-3}$. Furthermore, CSF movement is also *inter alia* induced by blood flow towards brain

causing pulsatility in its motion. The pulsatile component is ideally a linear combination of sinusoidal harmonics [40] given as

$$Flowrate = X + A\sin(\omega t + \alpha) + B\sin(2\omega t + \beta) \tag{2.6}$$

In equation (2.6), $\omega$ is the angular frequency and equals to $2\pi f$, where $f$ is the frequency of heartbeats per second. For simplicity, we assumed $f$ to be equal to 1 Hz. Whereas $X$ is bulk production of CSF and is approximately equal to 0.3 ml min$^{-1}$ ($6.25 \times 10^{-6}$ kg s$^{-1}$), A is equal to 0.21 ml min$^{-1}$ ($3.5 \times 10^{-6}$ kg s$^{-1}$) and B is equal to 0.05 ml min$^{-1}$ [40]. Phase difference $\alpha$ and $\beta$ are assumed to be zero because carotid artery, which takes blood away from heart, has zero phase difference [40].

It may also be clarified here that we have chosen specific points in lateral ventricles as inlets, because for modelling CSF flow (which is *per se* a creeping flow) this assumption is consistent with previous CSF flow modelling criteria [41–43].

# 3. Numerical method and calculation

The CSF flow modelling is done in ANSYS FLUENT wherein the equations of mass, momentum and energy conservation are solved (collectively called Navier–Stokes' equations). The fluid domain is discretized using finite volume formulation and at each cell centres following equations are solved [44]

$$\frac{\partial \rho}{\partial t} + \rho \nabla \cdot v = 0 \tag{3.1}$$

and

$$\rho\left(\frac{\partial v}{\partial t} + v \cdot \nabla v\right) = -\nabla P + \rho g + \mu \nabla^2 v \,, \tag{3.2}$$

where in equation (3.1), $\rho$ is the density of fluid, v is the velocity of the fluid and $\nabla$ is the gradient operator. This equation is called mass conservation (or continuity equation). While in equation (3.2), $\partial v/\partial t$ is the change in fluid velocity over time, $(v.\nabla v)$ is the convective acceleration, $\nabla P$ is the pressure gradient, $\rho g$ are external body forces (which may include gravity, electromagnetic forces effecting the fluid; for incompressible flows such as CSF the effect of gravity force on fluid motion is negligible) and $\mu \nabla^2 v$ is the viscous term which resists the motion of the fluid particles (can also be called internal stress).

Flow domain is solved using *viscous Laminar condition*. Pressure-based solver is used, with pressure–velocity coupling of PISO (pressure-implicit with splitting of operators). CSF pressures calculated from this step are using dynamically in the FEM model so that it represents a true picture. FEM model on the structural side contains large strain elements such as brain parenchyma; therefore, finite strain theory is applied to find element matrices and load vectors using the updated Lagrangian method as follows:

$$[\overline{K_i}]\Delta u_i = F_A - F_R, \tag{3.3}$$

where $F_A$ is the applied force and $F_R$ is the Newton–Raphson restoring force vector. $\overline{K_i}$ is given as follows:

$$[\overline{K_i}] = [K_i] + [S_i], \tag{3.4}$$

where $K_i$ is the stiffness matrix obtained through applying generalized stress–strain law of elasticity as follows:

$$[K_i] = [B_i]^T[D_i][B_i] \, dV, \tag{3.5}$$

where $B_i$ is the strain–displacement matrix in undeformed frame of reference $X$ and $D_i$ is the stress–strain matrix. While $S_i$ is the stress stiffness matrix given as follows:

$$[S_i] = [G_i]^T[\tau_i][G_i] \, dV, \tag{3.6}$$

where $G_i$ is the matrix shape functions, $\tau_i$ is the Kirchhoff stress tensor related to Cauchy stress tensor by Jacobian of the deformation gradient $J$ as given in equation (2.2). Newton–Raphson restoring force is given by

$$[F_i^{nr}] = \int [B_i]^T \sigma_i \, dV \tag{3.7}$$

Newmark integration scheme is used to solve all the above equations implicitly. Whereas Newton–Raphson method is used to converge loads, displacements and moments at each timestep and is given as

$$x_{n+1} = x_n - \frac{f(x_n)}{f'(x_n)}. \tag{3.8}$$

Implicit time formulation is used. Timestep size used in the instant study is taken to be 0.01 s. The specific value for timestep is calculated by ascertaining the auxiliary modal frequencies (using modal analysis) of the model and based on those frequencies we use the below formula

$$t = \frac{1}{20f}, \tag{3.9}$$

where $f$ is the maximum frequency required to be undertaken in the structural domain. Ideally speaking, any structure can have infinite number of natural frequencies as they are integral multiple of first natural frequency. However, it is a usual practice to take first six modal frequencies as those are the one which are often excited. The maximum frequency was taken (among six modes) and above equation was used to find out timestep size. This allows us to include those transient effects which may have a significant contribution towards deformation mechanics of brain. Whereas the total time was taken/assumed to be 1 s as standard simulation practice.

# 4. Results

The FEM model on the structural side has 769 322 elements, 678 149 nodes as of mesh density. In total, the simulation solved around 1 707 953 equations, apart from additional contact element formulations. The model is solved on a supercomputer with 28 computer nodes consisting of 12 core Intel Xeon E5-2620 2.0 GHz processors and 48 GB RAM on each node. Each model simulation completed in 24 h. Cumulatively it took 72 h for completion.

## 4.1. Deformation field and strains

In this section, we present results of deformation field. Figure 5 presents the deformation contour plot on axial, sagittal and coronal planes. The existing tumour is in nascent stage, and therefore, the deformation it produced on the brain parenchyma is less. Even the deformation progression in rest of the brain is minimal, suggesting that it is rather insignificant for practical terms. Figure 6 presents deformation volume rendering (in three-dimensions with skull obviously having zero deformation) for the case where tumour is grown by 16 pixels. Here the maximum deformation comes out to be 3.5 µm. The limits are scaled appropriately to visualize the results in three-dimensions. Figure 7 presents deformation contour (in three-dimensions) for the case where tumour is grown by 30 pixels. Here the maximum deformation comes out to be 4.52 µm. In all these cases, the deformation pattern, magnitude and its spread are different. As we shall see later, the associated ICP pressures created due to this tissue deformation also increases. In figure 5, the deformation produced is largely driven by an early growing tumour with a negligible associated magnitude. In figure 6, deformation increases twofold and its spread in its vicinity is larger compared with figure 5. Figure 7 depicts an even larger deformation spread and magnitude. The implications of this are discussed in the later section and contribute massively towards the translational merit of this study. Figure 8 shows a three-dimensional view of the deformation pattern for the case where tumour is grown 30 pixels. This figure also explains how deformation spread across the brain parenchyma and the reader can easily visualize the spread. Not only this; it depicts the relationship of tumour size with pressure distribution in the brain sac. To put into further context, figure 9 shows the strains across coronal, axial and sagittal sections; as they depict, the strain distribution inside cranium is significant and is generally well spread out. These strains contour plot show, and show quite clearly, that at some later point in tumour progression (post 30 pixels), deformation would continue to grow unabatedly. Figure 10 is an extension of earlier argument and shows deformation at each timestep in planes.

## 4.2. Stress field—intracranial cerebral pressure distribution

ICP distribution is the second most important variable which is studied in this study. We present stress results generated in response to the deformation produced above. We would like to emphasize here

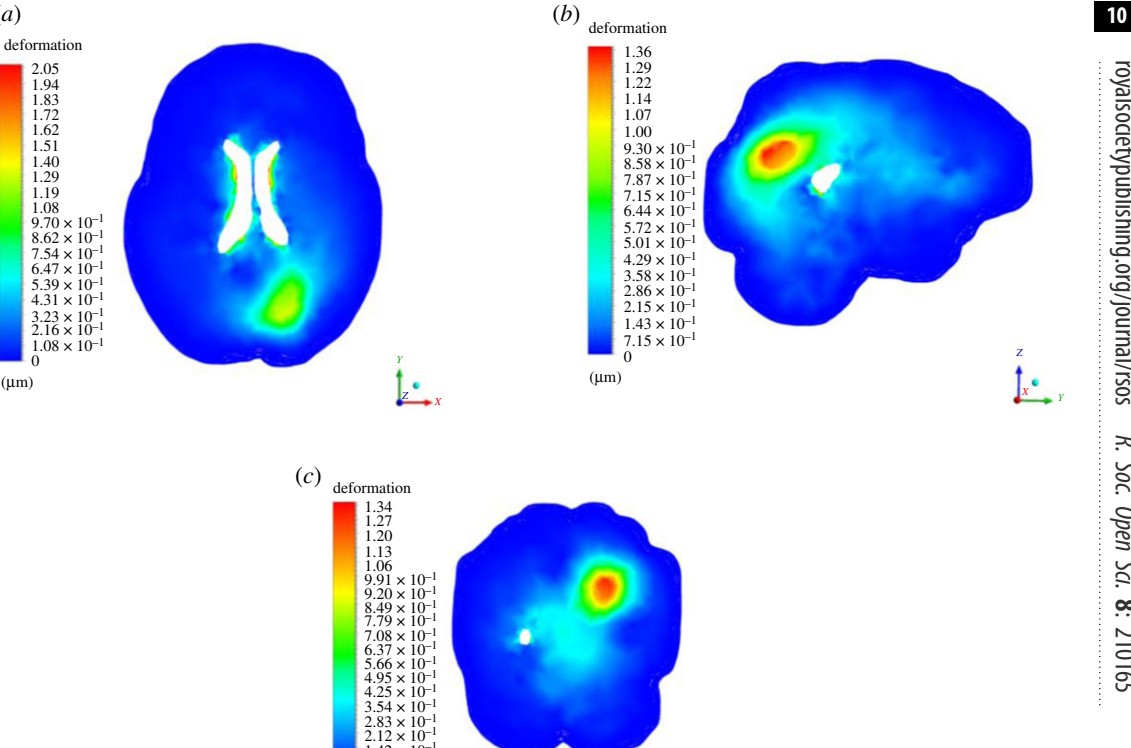

**Figure 5.** Deformation plots of axial, sagittal and coronal planes for existing tumour stage.

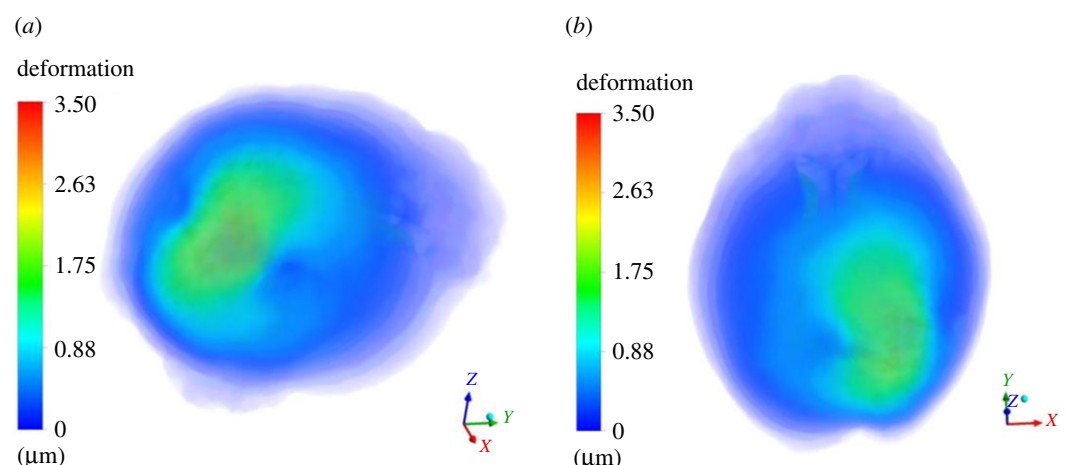

**Figure 6.** Three-dimensional contour plot of deformation on the entire brain parenchyma. The contour plot is designed in such way wherein skull and any other layer which has zero deformation is made to appear to be blue. The limits are appropriately scaled between 0 and 5 μm. Herein (a) shows the top view of the deformation plot; (b) shows the side view of the deformation plot.

further that our major focus of this study was to find out any correlation pattern between ICP magnitude and tumour case which was otherwise missing in the literature. Figure 11 shows volume rendering of von Mises stress for the case where tumour is grown by 16 pixels. The maximum stress generated is 615.25 Pa (4.61 mmHg), which is still within the normal range of ICP (5–15 mmHg). Whereas figure 12 shows the volume rendering of von Mises stress for the case where tumour is grown by 30 pixels. The maximum stress generated is 1488.73 Pa (11.16 mmHg), which again still is within the normal range of ICP (approx. 5–15 mmHg). The stress correlation in terms of quantitative assessment is significant. This explains some of the various mysteries attached to the brain tumour progression and poor prognosis if it is diagnosed at a later stage. However, suffice to say that even for a tumour grown uniformly by 30 pixels, the ICP value did not increase at level which goes beyond the normal range. In next section, we discuss the implications

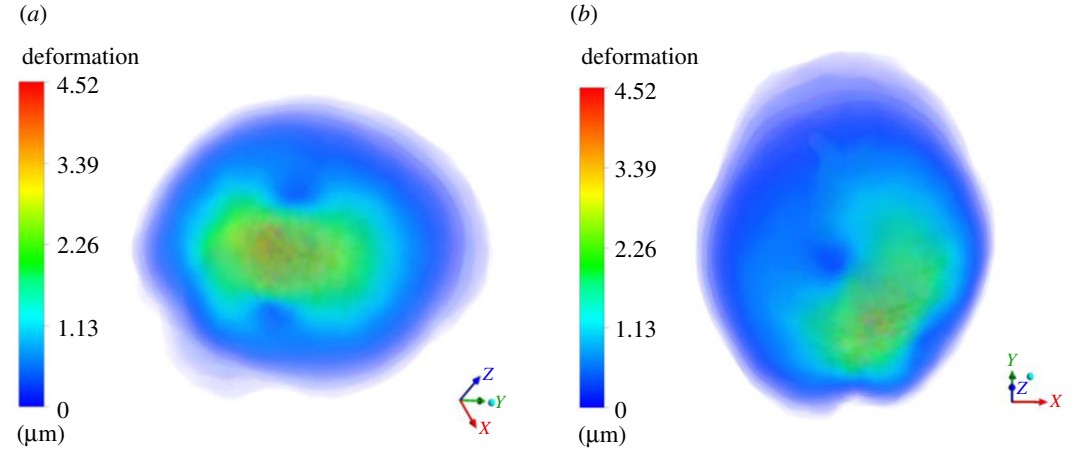

**Figure 7.** Three-dimensional contour plot of deformation on the entire brain parenchyma. The contour plot is designed in such way wherein skull and any other layer which has zero deformation is made to appear to be transparent. The limits are appropriately scaled between 0 and 5 μm. Herein (*a*) shows the top view of the deformation plot; (*b*) shows the side view of the deformation plot.

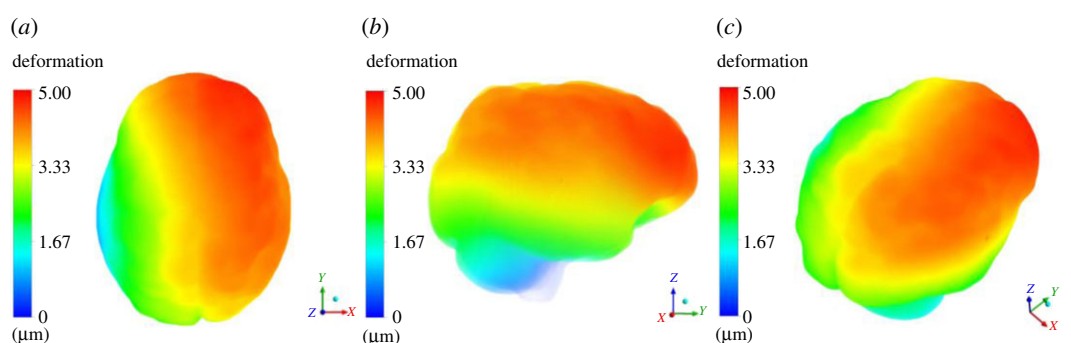

**Figure 8.** Three-dimensional contour plot of deformation on the entire brain parenchyma. The contour plot is designed in such way wherein skull and any other layer which has zero deformation is made to appear to be transparent. The limits are appropriately scaled between 0 and 5 μm. Herein (*a*) shows the top view of the deformation plot; (*b*) shows the side view of the deformation plot, whereas (*c*) shows a tilted view.

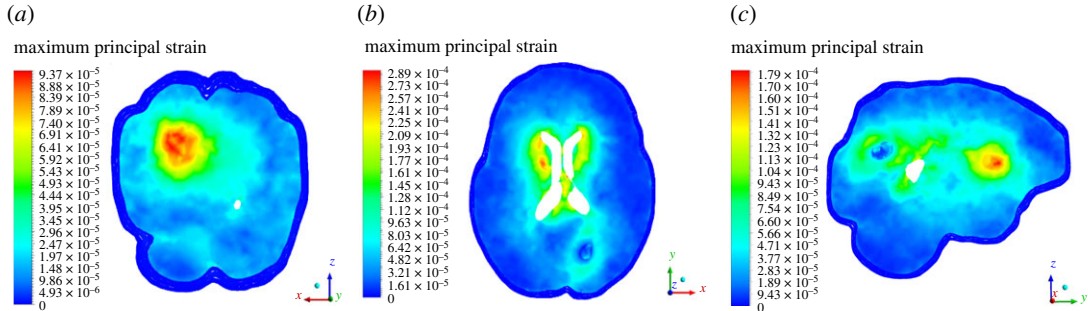

**Figure 9.** (*a*–*c*) Strain plot across coronal, axial and sagittal views. This strain plot is for the case when tumour is grown by 30 pixels.

of the obtained results and analyse whether they are meaningful or not and whether they contribute significantly in our understanding of intracranial dynamics under the influence of tumour.

## 5. Discussion

This paper is one of the first attempts in assessing quantitatively and objectively how brain tumour affects the intracranial environment, in particular, how much the deformation is produced when considering the effects of tumour forces. Dynamic pressures of CSF flow are also applied on the walls of ventricles using CFD analysis of CSF and fluid–structure interaction.

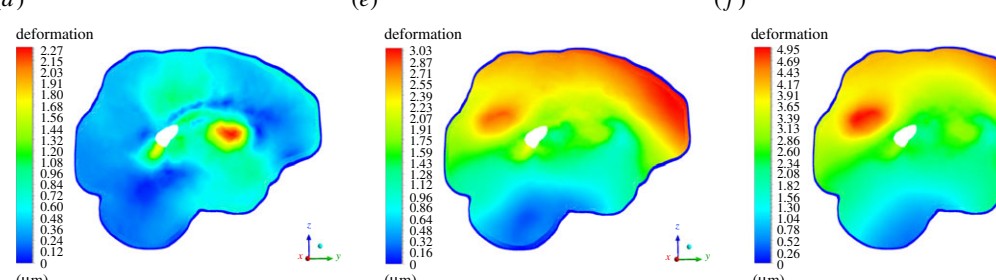

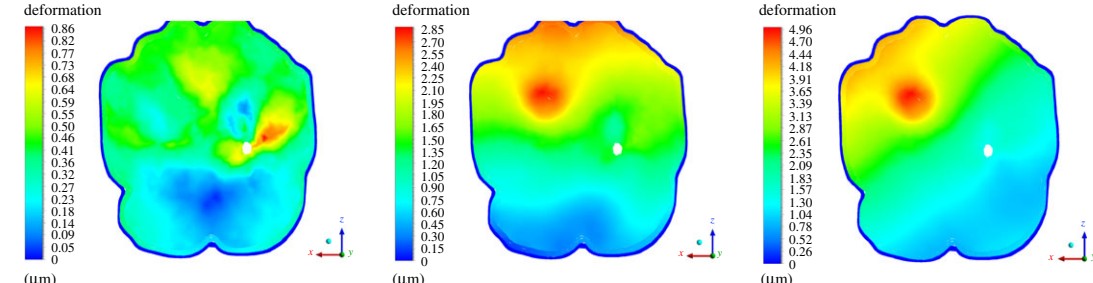

**Figure 10.** Deformation plot on axial, sagittal and coronal planes at different timestep (0.1, 0.5 and 1 s); (a–c) shows deformation contour plot on axial plane at 0.1, 0.5 and 1 s, respectively; (d–f) shows deformation contour plot on sagittal plane at 0.1, 0.5 and 1 s, respectively, whereas (g–i) shows deformation plot on coronal plane at 0.1, 0.5 and 1 s, respectively. Contours are scaled appropriately.

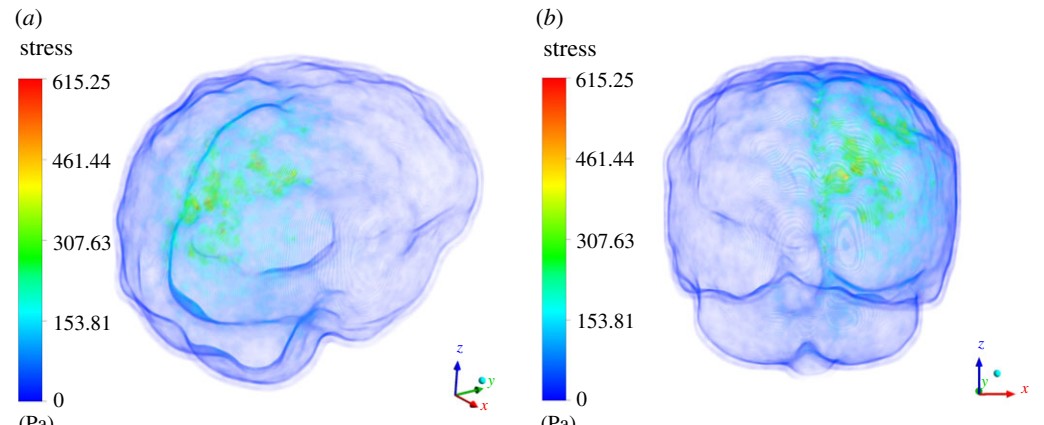

**Figure 11.** Three-dimensional volume rendering of the von Mises on the entire brain parenchyma. The contour plot is designed in such way wherein skull and any other layer which has zero stress is made to appear to be transparent. The limits are appropriately scaled. Herein (a) shows the top view of the stress plot; (b) shows the side view of the stress plot.

In this paper, real patient data is used, which strengthens the overall model because tumour spatial location is used in real time. To mechanically understand the effect of ballooning tumour, it is grown physically using voxel morphological operations so that the real-time geometry of tumour is provided in load application. Different brain layers are used in this study and, among them, brain parenchyma

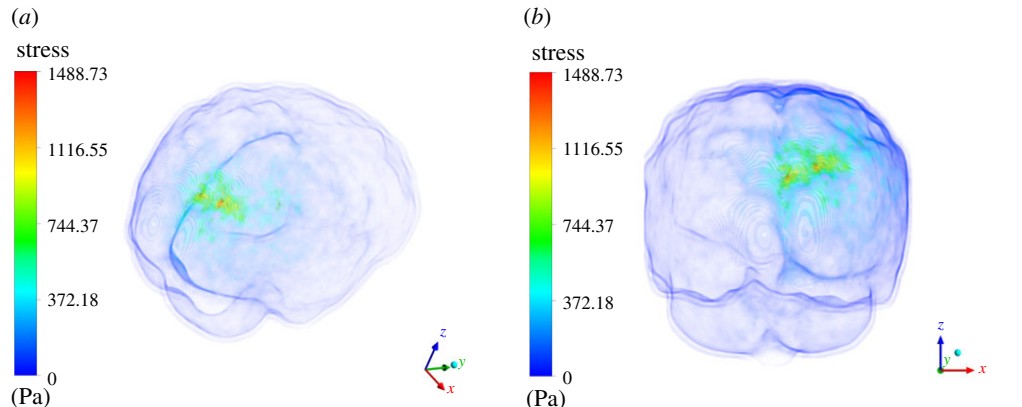

**Figure 12.** Three-dimensional volume rendering of the von Mises on the entire brain parenchyma. The contour plot is designed in such way wherein skull and any other layer which has zero stress is made to appear to be transparent. The limits are appropriately scaled. Herein (a) shows the top view of the stress plot; (b) shows the side view of the stress plot.

and ventricle body are of vital and pivotal importance. As a matter of fact, biological tissue has essentially both solid and fluid component [26]. Due to the presence of water, they are incompressible (or near incompressible). In this paper, we have modelled brain as hyper-viscoelastic because brain parenchyma has the tendency to produce large deformations if the pressures on the surface of parenchyma are relatively high, which can be seen in the results obtained herein above. Even in TBI cases, large deformations are produced due to sudden high impact load and, in them too, brain is modelled as hyper-viscoelastic (refer to [45]).

Three biomechanical models are developed. First is the normal case, wherein the real-time geometry of tumour is used, and resultant load is applied on the tumour uniformly in all radial directions. Second and third FEM models are essentially the same but have physically grown tumour of 16 and 30 voxels, respectively. Here too, load is applied uniformly by virtue of its size, uniformly in radial direction. The results obtained in this paper have categorically established in quantitative terms that

1. As tumour progresses in its shape and size the deformation pattern varies both in terms of magnitude and area of damage in vicinity.
2. The ICP correlation which was missing in literature has been addressed; even at tumour grown by 30 voxels uniformly the ICP value is still around 1488.73 Pa, which signals that even at normal ICP values (5–15 mmHg, approx. equal to max 2000 Pa) considerable deformation can occur inside brain.

This also explains why brain tumour is very hard to diagnose early on in terms of symptoms (such as headache etc.) since the required ICP value to produce such effects is still in evolution phase even though the physiological damage of brain is significant, and in some cases, plastic deformation can occur such as midline shift.

We would like to further emphasize that intracranial brain dynamics do not change in isolation. These pressures are studied when other factors such blood pressure, adequate CSF drainage in arachnoid villi etc. are normal and functioning well. This paper presupposes this assumption. And if these conditions change, the overall results may vary. However, before going into that realm of investigation, it is first essential to understand underlying ICP biomechanics fundamentally. Needless to say, the major focus of this paper is to first address this very area of knowledge and study the significant correlation between tumour size/growth with the ICP pressures, and to explain why early on poor prognosis is shown. This is exactly where this paper contributes significantly.

Furthermore, the results also explain why tumour in different brain regions produce different effects. From results of this paper, it can be safely implied and inferred that a tumour in brainstem region near to the cerebral aqueduct would quickly raise ICP values even higher since it would inevitably interact with the duct and can obstruct it. The resultant constriction of CSF may create internal imbalance resulting in higher CSF pressures on the ventricular body causing brain to expand uniformly in all directions causing hydrocephalus (and stenosis of cerebral aqueduct).

Even on temporal regions, ICP value may not rise substantially since tumour in that region can again deform the brain significantly (and mostly causing haematoma as well, refer to paper by [46], with a nominal increase in ICP). Furthermore, according to a paper [47], herniation due to haematoma can occur even at low ICP values. In the present study, we have also concluded on similar lines that even

for low ICP values significant brain deformation can occur. According to a statistically study [46], the adjusted normal range of ICP values is essentially (in his statistical study) in the range of 11.5 ± 4.5 mmHg. Hence even if this is provided then one can clearly see that the ICP values found out above are near to the cut-off range.

## 6. Limitations

Some limitations of the present study are briefly discussed here. First, in this study, we have not provided other neurological developments which may occur in the vicinity of tumour, as it grows physically. These neurological developments may include the development of cerebral oedema [48] or general disturbances in blood–brain barrier. This factor would be the next focus in our future studies pertaining to brain tumour. Second, while considering the application of tumour forces (and its growth), we assumed (and employed) that the tumour forces are uniformly applied in radial direction. However, it is only one of the scenarios (and phases) of tumour growth; there may be other growth stages where tumour may undergo and follow irregular growth trajectory. This aspect also needs further consideration in modelling brain tumour cases. Lastly, the growth of tumour employed in this study is based on voxel dilation technique (by 16 and 30 pixels mentioned above). It can certainly be argued that though this strategy is correct *per se*, but still it is an element of uncertainty in the possible growth of tumour exists—in its size, shape and volume. Since the temporal scope of this study was limited, therefore, real growth of tumour (in months and years) could not be obtained. Be that as it may, the prime focus of this study was to model tumour–ICP relationship and in future a thorough long-term study can be employed based on the model presented in this study.

## 7. Conclusion

This paper is one of the first attempts in assessing quantitatively and objectively how brain tumour affects the intracranial environment, in particular, how much the deformation is produced when considering the effects of tumour forces. The model presented in this study is non-invasive in nature and can be used in a medical setting to ascertain future likelihood scenarios about the state of deformation of the brain and ICP. We have also established that even at low ICP values, significant brain deformation can take place. We conclude, by way of inference, that since the external symptoms of tumour (such as headache etc.) are correlated to the magnitude of ICP, therefore, a low ICP value (as established above) may not trigger those symptoms, even at the scale of significant deformation. This then explains the dichotomy as to why symptoms of tumour come at a later stage.

Ethics. Patient Name: Ms. Naghmana Umair, Reference no. 00680–416. I confirm that I have been informed about the nature of the study and I have had opportunity to ask questions about the research. I voluntarily agree to share my MRI data with the researchers for this study. I understand that I can withdraw at any time without giving reasons and negative consequences. I agree to sign this consent form. Project title: Development and reconstruction of geometrically accurate models (3D) of Human Brain for calculation of ICP due to the presence of tumours Using Image-based modelling and Finite Elements. Grant: Higher Education Commission of Pakistan (9954).
Data accessibility. The data file is attached in the File upload section and also the patient consent form.
Authors' contributions. Z.M. supervised this research. Z.M. as principal investigator (PI) of this study conceived and conceptualized this research and mode of experimentation. Funding for this project was acquired by Z.M. M.U.U. and A.A. as research associates conducted all experiments, validation and analysis as well as paper writing. Z.M. and A.S. developed the theme storyboard and assisted in the overall organization of this paper. Z.M. reviewed and proofread. L.M. also provided assistance to review work.
Competing interests. We declare we have no competing interests.
Funding. This work is funded by Higher Education of Pakistan (HEC) under the auspices of National Research Program for Universities (NRPU) through an HEC grant code of 9954.
Acknowledgements. We are thankful to National University of Sciences and Technology (NUST) for providing Computer Labs and logistic support necessary to conduct this study.

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
