## [Peer Review File · Royal Society Open Science]

Review History

RSOS-210165.R0 (Original submission)

Review form: Reviewer 1 (B He)

Is the manuscript scientifically sound in its present form?

Yes

Are the interpretations and conclusions justified by the results?

Yes

Is the language acceptable?

Yes

Do you have any ethical concerns with this paper?

Yes

Have you any concerns about statistical analyses in this paper?

Yes

Recommendation?

Accept as is

Comments to the Author(s)

Real patient data is used which strengthens the overall model because tumor spatial location is used in real time. To mechanically understand the effect of ballooning tumor, it is grown physically using voxel morphological operations so that the real time geometry of tumor is catered in load application. Different brain layers are used in this study and, among them, brain parenchyma and ventricle body are of vital and pivotal importance.

Review form: Reviewer 2**Is the manuscript scientifically sound in its present form?**

Yes

Are the interpretations and conclusions justified by the results?

Yes

Is the language acceptable?

No

Do you have any ethical concerns with this paper?

No

Have you any concerns about statistical analyses in this paper?

No

Recommendation?

Major revision is needed (please make suggestions in comments)

Comments to the Author(s)

The general and specific comments on the manuscript are included in the attached file (see Appendix A).

Decision letter (RSOS-210165.R0)

Dear Mr UI Haq

The Editors assigned to your paper RSOS-210165 "How Growing Tumor Impacts Intracranial Pressure and Deformation Mechanics of Brain" have now received comments from reviewers and would like you to revise the paper in accordance with the reviewer comments and any comments from the Editors. Please note this decision does not guarantee eventual acceptance.

Please submit your revised manuscript and required files (see below) no later than 21 days from today's (ie 26-Jul-2021) date. Note: the ScholarOne system will 'lock' if submission of the revision is attempted 21 or more days after the deadline. If you do not think you will be able to meet this deadline please contact the editorial office immediately.

on behalf of Dr Derek Abbott (Associate Editor) and R. Kerry Rowe (Subject Editor)
openscience@royalsociety.org

Reviewer comments to Author:

Reviewer: 1

Comments to the Author(s)

Real patient data is used which strengthens the overall model because tumor spatial location is used in real time. To mechanically understand the effect of ballooning tumor, it is grown physically using voxel morphological operations so that the real time geometry of tumor is catered in load application. Different brain layers are used in this study and, among them, brain parenchyma and ventricle body are of vital and pivotal importance.

Reviewer: 2

Comments to the Author(s)

The general and specific comments on the manuscript are included in the attached file ("RSOS manuscript review_210165.pdf").

===PREPARING YOUR MANUSCRIPT===

===PREPARING YOUR REVISION IN SCHOLARONE===

- Any electronic supplementary material (ESM).
- If you are requesting a discretionary waiver for the article processing charge, the waiver form must be included at this step.
- If you are providing image files for potential cover images, please upload these at this step, and inform the editorial office you have done so. You must hold the copyright to any image provided.
- A copy of your point-by-point response to referees and Editors. This will expedite the preparation of your proof.

- Ensure that your data access statement meets the requirements at <https://royalsociety.org/journals/authors/author-guidelines/#data>. You should ensure that you cite the dataset in your reference list. If you have deposited data etc in the Dryad repository, please include both the 'For publication' link and 'For review' link at this stage.
- If you are requesting an article processing charge waiver, you must select the relevant waiver option (if requesting a discretionary waiver, the form should have been uploaded at Step 3 'File upload' above).
- If you have uploaded ESM files, please ensure you follow the guidance at <https://royalsociety.org/journals/authors/author-guidelines/#supplementary-material> to include a suitable title and informative caption. An example of appropriate titling and captioning may be found at https://figshare.com/articles/Table_S2_from_Is_there_a_trade-off_between_peak_performance_and_performance_breadth_across_temperatures_for_aerobic_scope_in_teleost_fishes_/3843624.

Author's Response to Decision Letter for (RSOS-210165.R0)

See Appendix B.

Decision letter (RSOS-210165.R1)

Dear Mr Ul Haq,

It is a pleasure to accept your manuscript entitled "How Growing Tumor Impacts Intracranial Pressure and Deformation Mechanics of Brain" in its current form for publication in Royal Society Open Science. The comments of the reviewer(s) who reviewed your manuscript are included at the foot of this letter.

Please ensure that you send to the editorial office an editable version of your accepted manuscript, and individual files for each figure and table included in your manuscript. You can send these in a zip folder if more convenient. Failure to provide these files may delay the

processing of your proof. You may disregard this request if you have already provided these files to the editorial office.

on behalf of Dr Derek Abbott (Associate Editor) and R. Kerry Rowe (Subject Editor)
openscience@royalsociety.org

Appendix A

Review Report

How Growing Tumor Impacts Intracranial Pressure and Deformation Mechanics of Brain

General Comments

The proposed work is focused on the modeling of tumor growth on changes in ICP. A patient based FEM model is created the tumor growth in the brain region is represented by voxel dilation technique. The deformation spread in the cerebral region is calculated and the strain, ICP values calculated. The article appears technically fine. The present study proposes some interesting facts related to early diagnosis of tumors, however certain points should be addressed before the manuscript can be published.

Specific Comments

1. Page 3, lines 37-40: The authors assume that the external symptoms of tumor are correlated primarily with ICP?. While I agree ICP is an important parameter here, there may be neurological developments in the tumor sites as it grows, and the diagnostics, specifically aimed at liquid biopsy protocols might still reveal vital info. This part needs to be elaborated and clarified.
2. Page 6, lines 43-46: Please discuss the possible limitations such technique to represent the expansion of tumors.
3. Page 11, line 38: Does the term on the left hand side of equation (5) represent strain energy density function?. Please check the analytical formulation for Ogden model presented in equation (5).
4. Page 12: lines 19-20: How is the value of 0.2 for the frictional coefficient justified? Please cite relevant literature.
5. Page 12, lines 22-24: Please justify the use of multipoint constrain formulation for brain parenchyma and ventricles.
6. Page 12, line 47: Explain the terms in the equations; for example 'w' should be circular frequency. Check other equations as well.
7. Page 13: Please check and correct the formulation for continuity equation presented in equation (10).

8. Equation (11) on page 13: The second term on right hand side and the physical relevance is not defined.
9. Page 14, lines 21-23: Please explain the physical considerations for choosing the time step and total time.
10. Page 15, Figure 5: The units of deformation is indicated as 'um'. The same pattern is followed in other relevant figures. It is unclear.
11. Page 18: Figure 12 (the vM stress corresponding to enlarged tumor) appears in half. A clear, complete figure is needed.
12. Page 19, line 52: "...resultant load is applied on the tumor uniformly in all radial directions" – I agree there are radial growth phases of tumors, but there can be focal emergence within a radial growth. Please discuss how such scenarios can be possibly modelled.
13. The manuscript language and presentation requires minor amendments; following are some samples (but not limited to) where corrections can be done:
 - (a) Page 3, line 37: 'From a diagnostic purpose' – purpose or perspective?
 - (b) Page 3, lines 41-43 – writing can be improved.
 - (c) Page 10, line 36: Typo, it should be 'fulfill' instead of 'full fill'.
14. Page 11, Figure 4 (a) and (b) appears differently formatted. Please use similar formatting for all figures.

Appendix B

Manuscript ID “RSOS-210165”

Paper Title: **How Growing Tumor Impacts Intracranial Pressure and Deformation Mechanics of Brain**

Order wise Author List: Ali Ahmed¹, Uzair Ul Haq¹, Zartasha Mustansar^{1*}, Lee Margetts² and Arslan Shaukat¹

Affiliation: ¹National university of Sciences & Technology (NUST), Islamabad 44000, Pakistan.

²Department of Mechanical, Aerospace and Civil Engineering, University of Manchester, UK

*Corresponding author Email: zmustansar@rcms.nust.edu.pk

We appreciate Journal of *Royal Society Open Science* editorial board for putting in their best efforts to find out the qualified reviewers and expedite the peer-review process. The editor and reviewers' comments are valuable. Based on reviewer's comments, the manuscript has been carefully revised by addressing all concerns point by point. Changes made in the manuscript are highlighted in green.

Response to Reviewer 1

There were no concerns raised by the learned reviewer on the content of the paper.

Response to Reviewer 2

At outset, we would like to appreciate learned reviewer for raising important and specific concerns on our study and we would like to address those concerns hereinbelow, point by point.

- 1) *Page 3, lines 37-40: The authors assume that the external symptoms of tumor are correlated primarily with ICP? While I agree ICP is an important parameter here, there may be neurological developments in the tumor sites as it grows, and the diagnostics, specifically aimed at liquid biopsy protocols might still reveal vital info. This part needs to be elaborated and clarified.*

Response: Agreed. Neurological development is an important thing to look at. Neurological development may also reveal significant clinical symptoms. Knowing that a metastatic brain tumors are usually diagnosed with a physical exam, a neurological exam, MRIs and other imaging techniques. Our study primarily focusses on information obtained from MRI scan alone and growing tumor using voxel dilation technique. We do agree that neurological development in the tumor vicinity may also provide a useful insight apart from the external symptoms alone. This could be a further step ahead to incorporate in our future studies pertaining to brain tumor.

For the second point, since our study holistically aims at computational modeling of brain tumor with respect to size, and most importantly using non-invasive approach. Therefore, from a diagnostic perspective, liquid biopsy might just help as one of the information files for this kind of study. In a liquid biopsy of tumor, a blood sample has to be taken from the target site to understand lethality of tumor and correlate with ICP mechanics. This may deviate from a non-invasive modeling perspective of this study. Apart from this, procedures including ICP measurement, where a catheter is inserted in tumor based-brain to measure ICP directly is also available- we still discourage this method by familiarizing a new perspective of non-invasive procedure by means of a good quality MRI scan.

- 2) *Page 6, lines 43-46: Please discuss the possible limitations such technique to represent the expansion of tumors.*

Response: A brief limitation sections is added. There we have addressed some limitations on the question raised above and also on some other which we think may have any direct relevance.

- 3) *Page 11, line 38: Does the term on the left hand side of equation (5) represent strain energy density function? Please check the analytical formulation for Ogden model presented in equation (5).*

Response: The equations (2), (3), (4) and (5) are now presented in a logical fashion. We would like to briefly explain here as well. A hyperelastic model presupposes a function called the strain energy density function (also called Helmholtz free energy per unit volume) denoted by φ , which is only dependent on the deformation gradient F of the body (Chaves, 2013). So effectively it can be said that $\varphi = \varphi(F)$. F can be calculated as follows (Lubarda, 2004; Vujošević, 2002):

$$F = \frac{\partial x(X,t)}{\partial t} \quad (2)$$

Further, the stress tensor and the Jacobian of the body can be calculated as follows:

$$J = \det [F] \quad (3)$$

$$\sigma = J^{-1} \frac{\partial \varphi(F)}{\partial F} F^T \quad (4)$$

where σ is the Cauchy stress tensor, φ is the strain energy density function, J is the Jacobian of the body as defined in equation (3). Equations (2), (3) and (4) are inter-related and once the strain energy density is calculated we can also calculate the Cauchy stress tensor σ . We now move towards calculation of strain energy density function, denoted by φ . The function φ is material behavior dependent. Usually, the strain energy density function is represented/formulated in terms of its principal stretches. Ogden Model (Ogden, 1972) is formulated in terms of principal invariants or Eigenvectors $\lambda_1, \lambda_2, \lambda_3$ and the associated strain energy behavior widely used in modeling biological tissues is given by:

$$\varphi = \varphi(\lambda_1, \lambda_2, \lambda_3) = \sum_{n=1}^N \frac{\mu_n}{\alpha_n} (\lambda_1^{\alpha_n}, \lambda_2^{\alpha_n}, \lambda_3^{\alpha_n} - 3) \quad (5)$$

Where N is the number of terms used in determining strain energy density function (such as Ogden 2 parameter or 3 parameter model), μ_n are material constants (shear moduli) and α_n are the experimental value (dimensionless).

- 4) *Page 12: lines 19-20: How is the value of 0.2 for the frictional coefficient justified? Please cite relevant literature.*

Response: The use of value of 0.2 for the frictional coefficient is used from a previous study. Reference is also added in the manuscript¹.

¹ Horgan, T. J., & Gilchrist, M. D. (2004). Influence of FE model variability in predicting brain motion and intracranial pressure changes in head impact simulations. *International Journal of Crashworthiness*, 9(4), 401-418

5) *Page 12, lines 22-24: Please justify the use of multipoint constrain formulation for brain parenchyma and ventricles.*

Response: Brain parenchyma is a 3D solid body (meshed by SOLID 187 elements) and ventricular body is a shell body (meshed by SHELL 181 element). The contact between them and the resultant assembly is called Shell-Solid assembly. To model contact between shell-solid assembly, Multipoint constraint formulation (MPC) is generally used². Further, the Multipoint constraint formulation (MPC) is a contact formulation which is particularly used for bonded contacts. Since, CSF pressures are applied on the ventricular body which expands its surface therefore contact formulation must be adequate enough to not only ensure that the contact remains intact (and hence penetration does not happen) but also convergence is achieved. The mechanics of this is as follows³. MPC internally adds constraint equations to “tie” the displacements between contacting surfaces. This approach is not penalty-based or Lagrange multiplier-based. It is a direct, efficient way of relating surfaces of contact regions which are bonded. Large-deformation effects also are supported with MPC-based bonded contact. Furthermore, rate of convergence is faster in MPC instead of Augmented Lagrange method. A brief justification is also added in the main manuscript with Table 2 presenting the justification for all contact formulation schemes (produced hereunder). Refer to table 2 in main manuscript.

Table 2. Contact Formulation Scheme

Tissue Connectivity	Type of Contact	Contact Formulation Scheme	Justification
Skull and Dura mater	Bonded	MPC	Preferred for bonded type contacts. Also Gives efficient convergence.
CSF and Brain parenchyma	Frictional ($\mu = 0.2$)	Augmented Lagrange Formulation	Preferred for Frictional contact schemes with large deformation problems.
Ventricles and Brain parenchyma	Bonded	MPC	Highly preferred for Shell-Solid body interface need efficient contact scheme where convergence difficulties can be overcome with no penetration.
Tumor and Brain parenchyma	Bonded	MPC	Preferred for bonded type contacts. Also Gives efficient convergence.
CSF and Dura Mater	No-Separation	Lagrange Multiplier Formulation	Since no-separation is used and at the same time penetration is to be avoided.

² For further reference, Chapter 7 ANSYS Mechanical APDL. "Contact technology guide." (2011)

³ Ibid

6) *Page 12, line 47: Explain the terms in the equations; for example, ‘w’ should be circular frequency. Check other equations as well.*

Response: Concern addressed. Terms of equations are completely defined, wherever appearing.

7) *Page 13: Please check and correct the formulation for continuity equation presented in equation (10).*

Response: Concern addressed. (Refer to equation 10)

8) *Equation (11) on page 13: The second term on right hand side and the physical relevance is not defined.*

Response: Concern addressed. (Refer to equation 11 and paragraph thereunder)

9) *Page 14, lines 21-23: Please explain the physical considerations for choosing the time step and total time.*

Response: First, the specific value for timestep is calculated by ascertaining the auxiliary modal frequencies (using modal analysis) of the model and based on those frequencies we use the below formula:

$$t = \frac{1}{20f}$$

Where f is the maximum frequency required to be undertaken in the structural domain. Ideally speaking, any structure can have infinite number of natural frequencies as they are integral multiple of first natural frequency. However, it is a usual practice to take first 6 modal frequencies as those are the one which are often excited, and so did we. The maximum frequency was taken (among 6 modes) and above equation was used to find out timestep size. This allows us to include those transient effects which may have significant contribution towards deformation mechanics of brain. Whereas, the total time was taken/assumed to be 1 second as standard simulation practice. A brief explanation is also added in main manuscript. Refer to last paragraph under the section Numerical Method and Calculation.

10) *Page 15, Figure 5: The units of deformation is indicated as ‘um’. The same pattern is followed in other relevant figures. It is unclear.*

Response: Primarily the magnitude of deformation of brain parenchyma is of the orders micrometers (10^{-6}). The legends of all graphs containing deformation have um as unit.

11) *Page 18: Figure 12 (the vM stress corresponding to enlarged tumor) appears in half. A clear, complete figure is needed.*

Response: Addressed. Refer figure 12.

12) *Page 19, line 52: “...resultant load is applied on the tumor uniformly in all radial directions” – I agree there are radial growth phases of tumors, but there can be focal emergence within a radial growth. Please discuss how such scenarios can be possibly modelled.*

Response: There can be various scenarios while modeling and applying tumor forces on the brain tissue. We considered one of the scenarios in which tumor forces were assumed to be uniformly applied in all radial directions. And one of the scenario can also be the focal emergence within tumor with a radial growth. This aspect can be considered in future studies.

13) *The manuscript language and presentation requires minor amendments; following are some samples (but not limited to) where corrections can be done:*

(a) *Page 3, line 37: ‘From a diagnostic purpose’ – purpose or perspective?*

(b) *Page 3, lines 41-43 – writing can be improved.*

(c) *Page 10, line 36: Typo, it should be ‘fulfill’ instead of ‘full fill’.*

Response: Concern addressed. Expression, grammar and writing style have been improved in the entire manuscript, including the points raised above, wherever necessary.

14) *Page 11, Figure 4 (a) and (b) appears differently formatted. Please use similar formatting for all figures.*

Response: Concern addressed. Figures are formatted uniformly. Figure 4 (b) is now better formatted wherein on y-axis we have log plot of stress against strain (on x-axis). This gives better visualization. Please Refer to figure 4 (a) and (b).